# Dimethyl Itaconate Reduces α-MSH-Induced Pigmentation via Modulation of AKT and p38 MAPK Signaling Pathways in B16F10 Mouse Melanoma Cells

**DOI:** 10.3390/molecules27134183

**Published:** 2022-06-29

**Authors:** Sungchan Jang, Won-Jae Chi, Seung-Young Kim

**Affiliations:** 1Department of Pharmaceutical Engineering & Biotechnology, Sun Moon University, Asan 31460, Korea; biochem1004@gmail.com; 2Microorganism Resources Division, National Institute of Biological Resources, Incheon 17058, Korea; wjchi76@korea.kr

**Keywords:** dimethyl itaconate, itaconic acid, GSK3β, anti-melanogenic, NRF2, MITF

## Abstract

Dimethyl itaconate (DMI) exhibits an anti-inflammatory effect. Activation of nuclear factor erythroid 2-related factor 2 (NRF2) is implicated in the inhibition of melanogenesis. Therefore, DMI and itaconic acid (ITA), classified as NRF2 activators, have potential uses in hyperpigmentation reduction. The activity of cyclic adenosine monophosphate (cAMP) response element-binding protein (CREB), an important transcription factor for MITF gene promoter, is regulated by glycogen synthase kinase 3β (GSK3β) and protein kinase A (PKA). Here, we investigated the inhibitory effect of ITA and DMI on alpha-melanocyte-stimulating hormone (α-MSH)-induced MITF expression and the modulatory role of protein kinase B (AKT) and GSK3β in melanogenesis in B16F10 mouse melanoma cells. These cells were incubated with α-MSH alone or in combination with ITA or DMI. Proteins were visualized and quantified using immunoblotting and densitometry. Compared to ITA, DMI treatment exhibited a better inhibitory effect on the α-MSH-induced expression of melanogenic proteins such as MITF. Our data indicate that DMI exerts its anti-melanogenic effect via modulation of the p38 mitogen-activated protein kinase (MAPK) and AKT signaling pathways. In conclusion, DMI may be an effective therapeutic agent for both inflammation and hyperpigmentation.

## 1. Introduction

The skin, the largest organ of the human body, provides protection against harmful external stresses, such as ultraviolet (UV) radiation and environmental pollutants [1,2]. Various cell types, including keratinocytes, melanocytes, fibroblasts, and skin-resident macrophage cells (Langerhans cells), mingle to form the skin [3]. Located in the outermost layer of the human skin, keratinocytes are mounted above melanocytes outnumbering them by approximately 10 to 1 [4]. Compared to keratinocytes, melanocytes are resistant to apoptosis owing to the presence of high levels of B-cell lymphoma 2 protein, which is also highly expressed in cancer cells [5,6]. Melanocytes proliferate slowly under normal conditions [4]. Microphthalmia-associated transcription factor (MITF) regulates melanin synthesis (melanogenesis) through timely and controlled expression of melanogenic genes such as tyrosinase (TYR) along with its own phosphorylation [7,8]. In response to stress such as UV radiation, keratinocytes secrete paracrine factors such as alpha-melanocyte-stimulating hormone (α-MSH) and stem cell factor (SCF). The binding of α-MSH and SCF to melanocortin-1-receptor (MC1R) and c-Kit, respectively, initiates signal transduction for melanogenesis [9,10]. The incoming signals from different receptors then converge to control the transcription and activity of MITF [8,11]. Melanin produced by melanocytes is eventually transferred to surrounding keratinocytes and exhibits characteristic pigmentation on the skin surface [12]. 

Keratinocytes, melanocytes, and Langerhans cells synthesize pro-opiomelanocortin and undergo proteolysis to produce α-MSH, β-MSH, and adrenocorticotropin (ACTH). Processed peptides bind to their own receptors on melanocytes (both α-MSH and ACTH on MC1R and β-MSH on MC4R) [10,13,14]. Activated MC1R in turn activates adenylyl cyclase (AC) to facilitate the conversion process of ATP to cyclic AMP (cAMP) [10,15]. The elevation in cAMP level leads to the activation of protein kinase A (PKA), which subsequently phosphorylates cAMP response element-binding protein (CREB). Transcription factors such as PAX3, SOX10, and CREB-regulated transcription coactivator (CRTC)-1, phosphorylated CREB binds to the promoter of the MITF gene and initiates its gene expression [15,16]. The expressed MITF proteins promote melanin synthesis by activating the transcription of melanogenesis-related genes such as TYR, TYR-related protein (TRP)-1, and TRP-2 [17,18,19]. Along with the activation of MC1R/AC/PKA/CREB signaling, α-MSH induces phosphorylation of β-catenin at Ser675 and activates protein kinase A (PKA), resulting in the attenuation of glycogen synthase kinase 3β (GSK3β) and the activation of Wnt/β-catenin signaling [20]. In addition, the phosphorylation of GSK3β at Ser9 by phosphoinositide 3-kinase (PI3K)-mediated protein kinase B (AKT) results in the stabilization and accumulation of β-catenin [20,21,22]. The stabilized β-catenin is then transported into the nucleus and enhances MITF transcription [20]. Multiple signaling pathways such as mitogen-activated protein kinase (MAPK) signaling pathways, c-Kit/Ras/Raf/MEK1/2/ERK/RSK, MC1R/AC/PKA/CREB, and Wnt/β-catenin converge on MITF to control the transcription and activity of MITF [9,10,11,20,21,22,23]. For example, the dual phosphorylation of MITF on Ser73 by ERK1/2 and Ser409 by p90 ribosomal S6 kinase potentiates transcription of MITF-dependent target genes [24,25]. The activation of CRTC3-3 by both MAPK/ERK and cAMP pathways results in increased melanin production and cell migration [26]. The SCF-induced phosphorylation of p38 stimulates CREB phosphorylation which subsequently induces the expression of MITF [27].

Nuclear factor erythroid 2-related factor 2 (NRF2) protects cells against oxidative stress by expressing anti-oxidation genes [28]. Under non-stressed conditions, NRF2 bound to Kelch-like ECH-associated protein 1 (KEAP-1) is subject to degradation by the ubiquitin-proteasome system [29]. Oxidative stress dislodges NRF from the KEAP1/NRF2 complex and restores the functionality of NRF2 as a transcription factor [28]. In addition to the involvement of NRF2 in anti-oxidation, its overexpression inhibits melanogenesis via the PI3K3/AKT/mammalian target of rapamycin (mTOR) signaling pathway in normal human melanocytes [30]. Moreover, NRF2 activators exhibit an anti-pigmentation effect in melanoma/melanocytes through the PI3K/AKT/mTOR/autophagy axis [30,31,32,33]. Both dimethyl itaconate (DMI) and itaconic acid (ITA) activate NRF2 by inactivating KEAP1. However, DMI shows better membrane permeability and strong electrophilicity than ITA [34,35,36,37]. While the anti-inflammatory effects of ITA and DMI have been well studied [34,35,36,37], the anti-melanogenic effects of these NRF2 activators have not been studied and this is the first to do so. In this study, we investigated the anti-melanogenic effects of ITA and DMI on α-MSH-induced MITF expression and the modulatory role of AKT and GSK3β in melanogenesis in B16F10 mouse melanoma cells. Our findings demonstrated that DMI exerted its depigmentation effect via modulation of AKT and p38 signaling pathways. 

## 2. Results 

### 2.1. Effect of ITA and DMI on α-MSH-Induced Melanin Production and Cell Viability

Melanin content and cell viability of the samples were measured as described in the Materials and Methods section. B16F10 melanoma cells were treated with three different concentrations (20, 40, and 80 μM) of ITA and DMI in the presence of α-MSH (200 nM) for three days. Relative to the untreated control, melanin content increased by approximately four-fold in the α-MSH-only sample. Melanin production was unaffected by treatment with ITA at all the tested concentrations, while DMI inhibited melanin production by 28, 36, and 42% at 20, 40, and 80 μM concentrations, respectively (Figure 1b). All samples showed >110% cell viability except for the 80 μM DMI-treated sample in which the cell viability was 97% (Figure 1c). These data indicated that ITA and DMI did not significantly affect cell viability at all the tested concentrations.

### 2.2. L-DOPA Oxidation 

Tyrosinase (TYR) functions as a rate-limiting enzyme in melanin production [38]. Oxidation of L-DOPA by intracellular cell lysate is frequently used to estimate TYR activity [39,40,41,42]. However, the studies by Schallreuter et al. [43], Land et al. [44], and Plonka et al. [45] reported that tyrosinases require L-DOPA for their own activation and may not produce dopaquinone via DOPA. We first investigated whether DMI exerted its inhibitory effect on α-MSH-induced melanin production through the inhibition of L-DOPA oxidation. B16F10 cells were treated with ITA and DMI at different concentrations (20, 40, and 80 μM), and melanin production was induced by adding α-MSH. Relative to the untreated control, α-MSH induced a 3.3-fold increase in L-DOPA oxidation. ITA treatment did not cause any statistically significant inhibition of L-DOPA oxidation (98, 100, and 99% L-DOPA oxidation at 20, 40, and 80 μM ITA, respectively) (Figure 2). In contrast, DMI treatment at 20, 40, and 80 μM concentrations inhibited α-MSH-induced L-DOPA oxidation by 0.5, 8, and 14%, respectively (Figure 2). Our data indicated that the melanin content was simultaneously decreased with the decrease in α-MSH-induced L-DOPA oxidation. 

### 2.3. Effect of ITA and DMI on the α-MSH-Induced Expression of Melanogenic Proteins

Melanin synthetic enzymes such as TYR, TRP-1, and dopachrome tautomerase (TRP-2) coordinate to produce melanin [9,10]. TYR, a copper-dependent metalloenzyme, catalyzes the oxidation of L-tyrosine to L-3,4-dihydroxyphenylalanine (L-DOPA) and dopaquinones [9,10]. B16F10 melanoma cells were treated with α-MSH (200 nM) in combination with either ITA or DMI and incubated for 48 and 72 h. The protein expression levels of MITF, TRP-1, and TRP-2 were normalized to those of β-actin and compared to those of α-MSH only. DMI suppressed α-MSH-induced MITF expression by approximately 8, 19, and 29% at 20, 40, and 80 μM concentrations, respectively, whereas ITA inhibited MITF expression by approximately 1, 6, and 12%, respectively, at the same concentrations (Figure 3a). The TYR induction by ITA was approximately 91, 83, and 59% at 20, 40, and 80 μM concentrations, respectively, while that by DMI was approximately 70, 46, and 46%, respectively, at the same concentrations (Figure 3b). The TRP-1 induction by ITA was approximately 91, 83, and 59% at 20, 40, and 80 μM concentrations, respectively, while that by DMI was approximately 78, 59, and 59%, respectively, at the same concentrations (Figure 3c). TRP-2 induction by ITA was approximately 80, 73, and 48% at 20, 40, and 80 μM concentrations, respectively, while that by DMI was approximately 60, 48, and 48% at 20, 40, and 80 μM, respectively (Figure 3d). Our data showed that DMI exhibited much stronger inhibitory effects on the α-MSH-induced expression of melanogenesis proteins such as MITF and TYR compared to ITA.

### 2.4. Time-Dependent Activation of AKT, GSK3β and MAPK Signaling Pathways in Response to α-MSH Treatment

According to the review by Cargnello and Roux, the three MAPK enzymes of extracellular signal-regulated kinase (ERK), c-Jun N-terminal kinase (JNK), and p38 regulate melanogenesis by relaying extracellular signals to intracellular responses [46]. For example, the binding of α-MSH to MC1R transactivates c-Kit/Ras/Raf/MEK1/2/ERK/RSK signaling [23]. In another instance, the phosphorylation of MITF at Ser73 and Ser409 by ERK and RSK enhances both transcriptional activity and degradation of MITF [24,25]. 

Previous studies showed the inhibitory role of the PI3K/AKT signaling pathway in melanogenesis. Shin et al. [30] suggested that NRF2 exerted an anti-melanogenic effect through the activation of PI3K/AKT/mTOR signaling in normal human epidermal melanocytes [30], Mosca et al. [47] demonstrated negative feedback on melanogenesis via the activation of the α-MSH-induced PI3K pathway [47], and Oka et al. [48] reported that PI3K inhibition increases melanin production while constitutively active mutant of AKT inhibits melanogenesis. 

NRF2 activators exert their hypopigmentation effects through the PI3K/AKT/mTOR/autophagy axis [30,31,32,33]. As per the review by Ho and Ganesan, mTOR normally acts to suppress autophagy and the depletion of mTOR causes the accumulation of MITF mRNA [33]. On the other hand, p-AKT inactivates GSK3β by phosphorylation at Ser9 [49], leading to the enhanced transcription of MITF. In addition, the studies by Grime and Jope [50] and Bellei et al. [20,49] demonstrated that GSK3β inhibits CREB DNA binding activity while GSK3β inhibition promotes melanogenesis in B16 melanoma. Taken together, these studies showed the inhibitory role of mTOR and GSK3β in melanogenesis.

Due to the significant involvement of MAPK, AKT, and GSK3β signaling pathways in the α-MSH-induced transcription of MITF and melanogenesis, we investigated the time-dependent phosphorylation of MAPK proteins, AKT, and GSKβ in B16F10 melanoma cells at five time points (0, 15, 30, 60, and 120 min) after α-MSH treatment. When normalized to the expression level of phosphorylated ERK (p-ERK) to that of total ERK and compared to the basal expression level at time zero, the relative phosphorylation levels of ERK were approximately 568, 542, 347, and 243% at 15, 30, 60, and 120 min after α-MSH stimulus, respectively (Figure 4a). The phosphorylation of p38 induces phosphorylation of CREB which in turn stimulates the expression of MITF [27]. When normalized to that of p38 and compared to the basal expression level at time zero, the relative phosphorylation levels of p38 were approximately 347, 284, 258, and 145% at 15, 30, 60, and 120 min after α-MSH treatment, respectively (Figure 4b). The regulatory role of JNK in melanogenesis remains unclear. Kim et al. reported the forskolin-induced inhibition of JNK in Mel-Ab mouse melanocytes, while the study by Han et al. showed the α-MSH-induced activation of JNK in B16F10 melanoma cells [26,51]. The time-course analysis of the phosphorylation of JNK showed that the phosphorylation level of JNK relative to that at time zero was increased by approximately 5% at 15 min post-α-MSH treatment but decreased to approximately 82, 42, and 51% at 30, 60, and 120 min, respectively (Figure 4c). 

In addition to α-MSH-triggered activation of MAPK proteins, we investigated the time-dependent phosphorylation of AKT and GSK in response to α-MSH. The phosphorylation level of AKT at each time point was normalized to that of total AKT and compared to that at time zero. In line with the research by Mosca et al. [47], the expression level of p-AKT decreased by approximately 45 and 63% at 15 and 30 min after α-MSH treatment, respectively (Figure 4d). However, the relative phosphorylation level of AKT at 1 h bounced back to that at time zero and further increased to approximately 130% at 2 h after α-MSH treatment (Figure 4d). The time-dependent increase or decrease in GSK3β phosphorylation following α-MSH treatment was similar to that of AKT (Figure 4d,e). When normalized to β-actin, the α-MSH-induced phosphorylation level of GSKβ relative to that at time zero was approximately 59, 70, 100, and 110% at 15, 30, 60, and 120 min, respectively, while the protein expression level of GSK3β relative to a zero point was approximately 79, 89, 100, and 115% at their respective time points (Figure 4e). When compared to those at time zero, the relative expression levels of p-GSKβ to GSKβ were approximately 75, 78, 100, and 96% at 15, 30, 60, and 120 min after α-MSH treatment (Figure 4e).

The study by Kim et al. [52] indicated that the inhibition of AKT by α-MSH is concomitant with the down-regulation of phosphorylated mTOR (p-mTOR). In addition, the study by Hah et al. showed that mTOR inhibition by rapamycin induces melanogenesis in human MNT-1 melanoma cells [32]. Taken together, these studies indicated that α-MSH treatment inhibits the activation of AKT/mTOR signaling and induces melanogenesis. In addition to the inhibitory role of GSK3β in the DNA binding activity of CREB, p-AKT-mediated GSK3β inhibition promotes melanin production [49]. In agreement with the results of the aforementioned studies, our results showed that after α-MSH treatment there was a short period of time within which the α-MSH-induced expression level of p-AKT, p-GSK3β, and GSK3β was below those at time zero (Figure 4d,e), which may contribute to α-MSH-induced melanogenesis. 

MAPK, AKT, and GSK3β signaling pathways converge to initiate MITF transcription [10,20]. When normalized to that of β-actin and compared to the basal expression level at time zero, the relative expression level of MITF increased by 1.11-, 1.26-, 1.33-, and 1.9-fold at 15, 30, 60, and 120 min after α-MSH treatment, respectively (Figure 4f). The relative expression level of MITF continued to increase over the tested time points.

### 2.5. Effect of ITA and DMI on the α-MSH-Induced Activation of MAPK, AKT, and GSK3β Signlaing Pathways 

From the time-course experiment, it was found that 15 min after α-MSH treatment, the phosphorylation of MAPKs peaked although the phosphorylation of AKT was maximally inhibited at 30 min after α-MSH treatment. Therefore, this time point (15 min) was used to assess the effect of ITA and DMI on α-MSH-induced phosphorylation of MAPKs (ERK, JNK, and p38) and AKT. The band intensities for phosphorylated proteins were normalized to their respective total protein levels. The band intensities for MITF, p-GSK3β, and GSK3β were normalized to β-actin. The phosphorylation levels of MAPK, MITF, p-GSK3β, and GSK3β were compared with those of α-MSH only. 

Relative to the untreated sample, α-MSH increased ERK phosphorylation by approximately four-fold. Relative to the α-MSH-only sample, the phosphorylation levels in the co-treatment samples of 20, 40, and 80 μM of ITA with α-MSH were approximately 100, 105, and 106%, respectively, while those of DMI with α-MSH were approximately 100, 109, and 101%, respectively, at the same concentrations (Figure 5a). The JNK phosphorylation levels of ITA in the presence of α-MSH were approximately 78, 95, and 88% at 20, 40, and 80 μM, respectively, while those of DMI in the presence of α-MSH were 84, 81, and 90%, respectively, at the same concentrations (Figure 5b). Treatment with 20, 40, and 80 μM DMI inhibited α-MSH-induced p38 phosphorylation by approximately 0, 16, and 35%, respectively. In contrast, ITA at 20, 40, and 80 μM concentrations increased α-MSH-induced p38 phosphorylation by approximately 42, 19, and 24%, respectively (Figure 5c). The p38-mediated phosphorylation of CREB stimulates the transcription of MITF [27]. The data indicate that DMI may inhibit α-MSH-induced pigmentation via modulation of p38 signaling. 

In addition to MAPK signaling, we examined the depigmentation effect of ITA and DMI through PI3K/AKT/mTOR axis and GSK3β signaling in α-MSH-treated B16 melanoma cells. As expected, at 15 min after the α-MSH treatment, α-MSH suppressed p-AKT expression. Our data showed that DMI at 40 and 80 μM concentrations exerted a better stimulatory effect on p-AKT expression relative to both α-MSH-only treatment as well as α-MSH and ITA co-treatments (Figure 6a). Neither ITA nor DMI has a statistically significant effect on the α-MSH-induced expression of p-GSK3β and GSK3β (Figure 6b).

NRF2 activators exhibit their hypopigmentation effects via modulation of the PI3K/AKT/mTOR/autophagy axis [30,31,32,33]. p-AKT inactivates GSK3β by phosphorylation at Ser9 [49], leading to the enhanced transcription of MITF. In addition, GSK3β inhibits CREB DNA binding activity while GSK3β inhibition promotes melanogenesis in B16 melanoma and normal human melanocytes [49,50]. Taken together, the activation of mTOR and GSK3β signaling pathways are associated with the inhibition of melanogenesis.

To further evaluate the depigmentation effect of ITA and DMI via α-MSH-activated AKT and GSK3β signaling pathways, B16F10 cells were cultured in the presence or absence of α-MSH in combination with ITA or DMI (40 μM) for 6 h. The expression level of p-AKT was normalized to that of total AKT. The expression levels of p-GSK3β, GSK3β, and MITF were normalized to those of β-actin. The relative expression level of each protein was then compared to the basal expression level in untreated or α-MSH-stimulated cells. Compared to the untreated control cells, the treatment of ITA and DMI either alone or in combination with α-MSH increased the relative expression levels of p-AKT, p-GSK3β, and GSK3β (Figure 7a–c). However, when compared to the co-treatment group of ITA and α-MSH, the co-treatment group of DMI and α-MSH showed stronger and weaker inhibitory effects on the α-MSH-induced expression of MITF and p-AKT, respectively (Figure 7a,d). Our data showed that the increased expression of p-GSK3β and GSK3β correlated positively with the increased expression of p-AKT (Figure 7a,b,c). The results thus indicated that DMI at 40 μM exerted its inhibitory effect on α-MSH-induced MITF expression via modulation of AKT and GSK3β signaling pathways.

## 3. Discussion

NRF2 activators such as ITA and DMI provide cells with protection against oxidative stress. In addition, it is suggested that the overexpression of NRF2 may inhibit melanogenesis through the PI3K3/AKT/mTOR axis/autophagy axis [30,31,32,33]. While the anti-inflammatory properties of ITA and DMI have been well examined [34,35,36,37], the depigmentation effects of these two NRF2 activators have not been studied and this is the first to do so. The expression and activity of MITF are modulated through MAPK, AKT, and GSK3β signaling pathways [11,19,20,46,47,48,49,50]. Our time-course analysis of the phosphorylation of AKT and GSK3β signaling molecules in response to α-MSH treatment showed that there was a brief period of time within which the α-MSH-induced expression of p-AKT, p-GSK3β, and GSK3β was below that at time zero and this may enhance MITF transcription and its activity. In the follow-up experiments, the inhibitory effect of DMI and ITA on the α-MSH-induced expression of melanogenesis enzymes and the regulatory role of AKT and GSK3β in melanogenesis were assessed. Our data showed that DMI down-regulated the α-MSH-induced expression of melanin synthetic enzymes such as MITF in a concentration-dependent manner in B16F10 cells via modulation of AKT and p38 signaling pathways. DMI exhibited visibly evident depigmentation effects in α-MSH-treated B16F10 melanoma cells, whereas ITA did not. The results of this study are in line with those by Kim et al. [52] that DMI exerted its anti-melanogenic effect by up-regulation of p-AKT and by down-regulation of p38 in B16F10 melanoma cells. As reported by Smalley and Eisen [53], Bellei et al. [54], and Kim et al. [55], our results suggested pigmentation through p38 activation. However, p38 silencing ultimately increases melanin production [54]. Thus, further research on the role of p38 in melanogenesis is required. In addition, L-DOPA oxidation assay alone would not provide accurate measurements for tyrosinase activity [43,44,45]. For this reason, the combined use of L-DOPA oxidation assay and other tyrosinase assay methods such as mushroom tyrosinase-based enzyme inhibition assay and tyrosinase zymography [56] would confirm the anti-tyrosinase activities of DMI. In conclusion, DMI may be an effective therapeutic agent for both inflammation and hyperpigmentation.

## 4. Materials and Methods

### 4.1. Chemicals and Equipments

B16F10 mouse melanoma cells (cat. no. CRL-6745) were purchased from the American Type Culture Collection (ATCC) (Manassas, VA, USA). Phenol red-free Dulbecco’s Modified Eagle’s Medium (DMEM; cat. no. LM001-10), sodium pyruvate (NaPy; cat. no. LS013-01), penicillin-streptomycin (P/S; cat. no. LS202-02), and fetal bovine serum (FBS; cat. no. S001-01) were supplied by Welgene Inc. (Gyeongsan-si, Gyeongsanbuk-do, Republic of Korea). Phosphate-buffered saline (PBS; cat. no. PR2004-100-72) and radioimmunoprecipitation (RIPA) lysis buffer (cat. no. R2002) were purchased from Biosesang (Seongnam-si, Gyeonggi-do, Republic of Korea). ITA (A15566; Alfa Aesar, Heysham, England). DMI (I0206; Tokyo Chemical Industry, Tokyo, Japan). Primary antibodies from Cell Signaling Technology (Danvers, MA, USA) were used for immunoblotting: MITF (D5G7V; cat. no. 1259S), P-p44/42 MAPK (T202/Y204; cat. no. 9101S), p44/42 MAPK (ERK1/2; cat. no. 9102S), P-p38 (T180/Y182) (cat. No. 9211S), p38 (cat. no. 9212S), p-SAPK/JNK (T183/Y185; cat. no. 9251S), SAPK/JNK (cat. no. 9252S), p-AKT (S473; cat. no. 9271S), AKT (cat. no. 9272S), Phospho-GSK-3-beta (Ser9; cat. no. 9322), GSK-3-beta (cat. no. 9315). Beta-actin (cat. no. VMA00048) primary antibody, ClarityTM Western ECL substrate (cat. no. 170-5060), Trans-Blot Turbo RTA Transfer Kit (cat. no. 170-7272), and the Trans-Blot TurboTM Transfer System (cat. no. 170-4150) were obtained from Bio-Rad (Hercules, CA, USA). NRF2 (H-10; cat. no.518036) primary antibody was purchased from Santa Cruz Biotechnology (Dallas, TX, USA). L-Dopa-(phenyl-d3) (L-DOPA) (cat. no. 333786) and protease inhibitor cocktail (PI) (cat. no. P8340) were supplied by Sigma-Aldrich (St. Louis, MO, USA). Falcon 96-well clear flat-bottom TC-treated culture microplate (cat. no. 353072; Corning, Glendale, AZ, USA). Pierce BCA Protein Assay Kit (cat. no. 23225; Thermo Scientific, Rockford, IL, USA). Image Reader (LAS 4000 Mini; Fujifilm, Tokyo, Japan). Microplate reader (MultiSkan Go) (Thermo Scientific, Rockford, IL, USA). Tween-20 (cat. no. T0886) was bought from Samchun (Pyeongtaek-si, Gyeonggi Province, South Korea). Tris-buffered saline with Tween-20 (1X TBST) was prepared by mixing 1 mL of Tween-20 with 1 L of tris-buffered saline (24.2 g of tris, 8 g of NaCl, pH 7.6).

### 4.2. Cell Culture

B16 culture medium was composed of 450 mL of DMEM, 50 mL of heat-inactivated (at 56 °C for 30 min) FBS, and 5 mL of NaPy. B16F10 cells (2.5 × 10^5^ cells) were plated in 10 mL culture medium. Sub-cultivation was performed every two days by washing once with PBS and a brief rinse with trypsin-EDTA. Trypsin-EDTA was then removed from the plate and incubated at 37 °C and 5% CO_2_. Once the cell layer was dispersed, the cells were collected in the B16 culture medium and centrifuged at 1000 RPM for 3 min. The resulting pellet was resuspended in B16 culture medium, and the cell number was counted.

### 4.3. MTT Assay

The cells (2 × 10^3^ cells) were seeded in 200 μL of culture medium per well in a 96-well plate and cultured at 37 °C and 5% CO_2_ for 18 h. Further, the cells were treated with α-MSH alone or in combination with three different concentrations (20, 40, and 80 μM) of ITA and DMI. After incubation for 48 or 72 h at 37 °C and 5% CO_2_, the culture medium was replaced with B16 medium containing 0.6 mg/mL 3-(4,5-dimehtylthiazol-2-yl)-2,5-diphenyltetrazolium bromide (MTT) and incubated for 1 h at room temperature. DMSO (200 μL per well) was added to the resulting formazan crystals which were then put on an orbital shaker at 200 rpm for 5 min. The color intensities of the plates were measured at an absorbance wavelength of 590 nm. The obtained absorbance values were compared to those of the α-MSH-only samples and converted to percentages.

### 4.4. Melanin Quantification 

After seeding in a 96-well plate as described above, the cells were treated with α-MSH alone or in combination with ITA or DMI for 72 h. The absorbance of melanin was measured at 405 nm wavelength. The obtained absorbance values were compared to those of the α-MSH-only samples and converted to percentages. 

### 4.5. L-DOPA Oxidation

L-DOPA oxidation was measured as previously described methods with slight modifications [39,40,41,42]. B16F10 cells (5 × 10^4^ cells) were seeded at 2 mL per well in two 6-well plates and incubated at 37 °C and 5% CO_2_ for 18 h. The cells were then treated with α-MSH (200 nM) alone or with three different concentrations (20, 40, and 80 μM) of ITA or DMI and further incubated for 72 h at 37 °C and 5% CO_2_. After one wash in PBS, the cells were lysed in protease inhibitor cocktail-added RIPA buffer and mixed on a rocking shaker for 2 h at 4 °C. The cell lysate was then collected in a 2 mL tube and centrifuged in a refrigerated benchtop centrifuge at 12,000 rpm for 30 min. The supernatant was transferred to a new tube, and the proteins in the tube were quantified using bicinchoninic acid (BCA) assay. L-DOPA powder was solubilized in 0.1 M sodium phosphate buffer (pH 6.8) at a final concentration of 2 mg/mL. The samples were diluted to 1 μg/μL using the previously used PI-added 1X RIPA buffer. Cell lysate (20 μL) and diluted L-DOPA (80 μL) were added to the wells of a 96-well microplate and wrapped in aluminum foil. Following a one-min shaking on an orbital shaker, the plate was incubated at 37 °C until the color difference between the untreated control and sample treatments became clear. The absorbance values were measured at 490 nm using a microplate reader [57]. 

### 4.6. Immunoblotting

Equal amounts of protein were loaded onto each well in a gel (8% resolving gel, 4% stacking gel), and the proteins were separated by molecular weight. The proteins on the gel were then transferred to a polyvinylidene difluoride membrane using the Trans-Blot Turbo RTA Transfer Kit and Trans-Blot Turbo Transfer System (2.5 A constant, up to 25 V, 13 min). The blot was washed once with 1X TBST and blocked with 5% w/v skim milk-added 1X TBST buffer for 1 h. After washing thrice for 10 min each, the blot was incubated in primary antibody buffer for 18 h on a shaker at 4 °C. After collecting the primary antibody buffer for reuse, the blot was washed twice for 10 min each and incubated in secondary antibody buffer with shaking at room temperature for 1 h. The blots were subsequently washed three times for 10 min each. Further, the proteins present in the blots were visualized using Western ECL substrates and a LAS 4000 MINI Image Reader. Immunoblot bands were quantified using the ImageJ software (NIH, United States). The band intensities for phosphorylated proteins were normalized to their respective total protein levels. The band intensities for MITF, p-GSK3β, and GSK3β were normalized to β-actin

### 4.7. Statistical Analysis

Statistical analysis was performed using Microsoft Excel 2010. Data were reported as the mean ± standard deviation of three independent experiments. Statistical differences between the means of sample groups were resolved by Student’s *t*-test. * *p* < 0.05, ** *p* < 0.01, *** *p* < 0.005, **** *p* < 0.001, ^#^ *p* < 0.05, ^##^ *p* < 0.05.

## Figures and Tables

**Figure 1 molecules-27-04183-f001:**
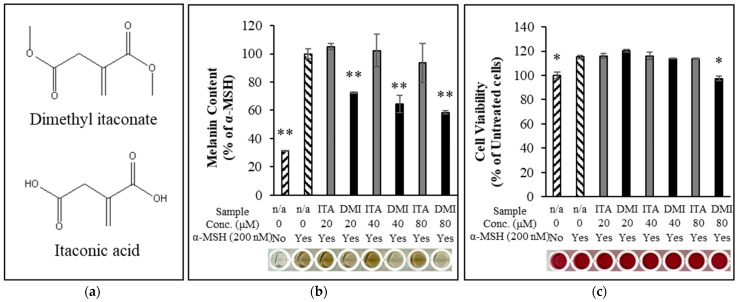
(**a**) Chemical structure of dimethyl itaconate (top) and itaconic acid (bottom). (**b**) Dimethyl itaconate showed a significant reduction in α-MSH-induced melanin production relative to itaconate acid. (**c**) The cell viability for all treatment samples was over 110% except 80 μM DMI at which the cell viability was about 97%. The data were presented as the mean ± standard deviation of three independent experiments; * *p* < 0.05 and ** *p* < 0.01 compared with α-MSH only and untreated cells for melanin assay and cell viability assay, respectively.

**Figure 2 molecules-27-04183-f002:**
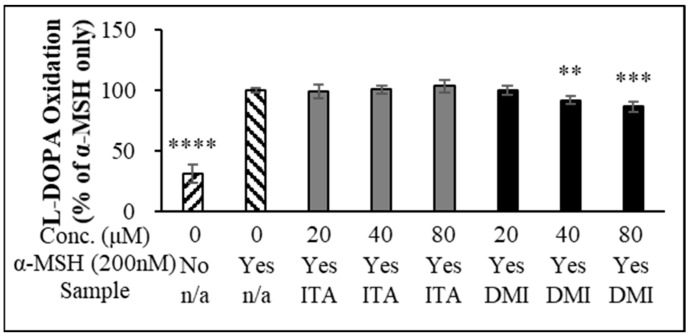
L-DOPA oxidation. The data were presented as the mean ± standard deviation of three independent experiments; ** *p* < 0.01, *** *p* < 0.005, **** *p* < 0.001 compared with α-MSH only.

**Figure 3 molecules-27-04183-f003:**
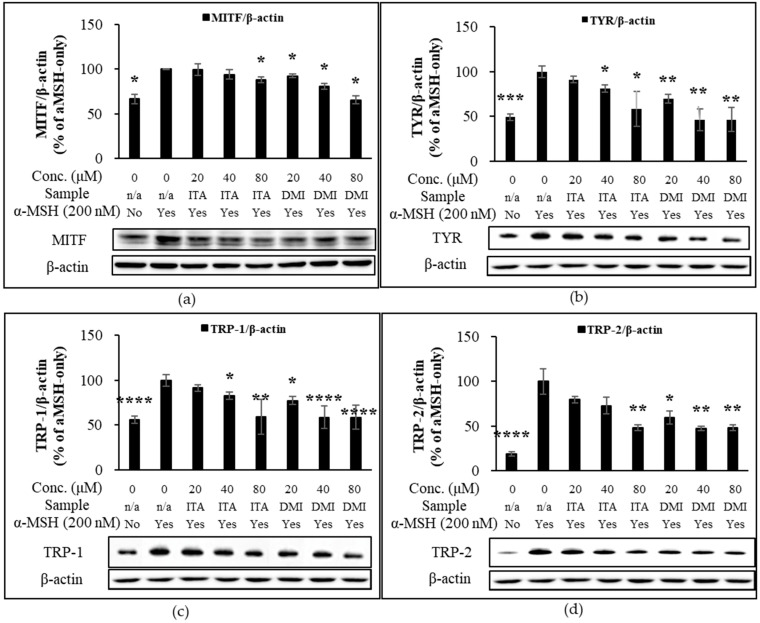
Effect of ITA and DMI on the α-MSH-induced expression of melanogenesis proteins. The expression level of melanogenic proteins was normalized to that of β-actin and compared to that of α-MSH only. (**a**) MITF/β-actin. (**b**) TYR/β-actin. (**c**) TRP-1/β-actin. (**d**) TRP-2/β-actin. The data were presented as the mean ± standard deviation of three independent experiments; * *p* < 0.05, ** *p* < 0.01, *** *p* < 0.005, **** *p* < 0.001 compared with α-MSH only.

**Figure 4 molecules-27-04183-f004:**
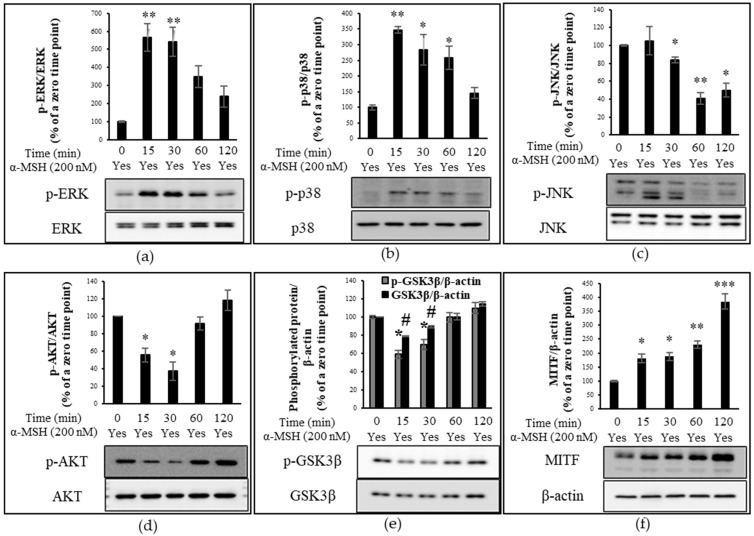
Proteins detected by immunoblotting. Time-course analysis of protein phosphorylation level (percentage ratio of phosphorylated protein to total protein) or protein expression level (percentage ratio of total protein to β-actin) in α-MSH-treated B16F10 cells. The relative phosphorylation level of extracellular signal-regulated kinase (ERK), p38, and JNK as well as the protein expression level of microphthalmia-associated transcription factor (MITF), glycogen synthase kinase 3β (GSK3β), and phosphorylated-GSK3β (p-GSK3β) are compared to those at time zero (control). (**a**) The relative phosphorylation of ERK. (**b**) The relative phosphorylation level of p38. (**c**) The relative phosphorylation level of JNK. (**d**) The relative phosphorylation level of AKT. (**e**) The expression level of p-GSK3β and GSK3β. (**f**) The relative protein expression of MITF. The data were presented as the mean ± standard deviation of three independent experiments; * *p* < 0.05, ** *p* < 0.01, *** *p* < 0.005, ^#^ *p* < 0.05 compared with α-MSH only.

**Figure 5 molecules-27-04183-f005:**
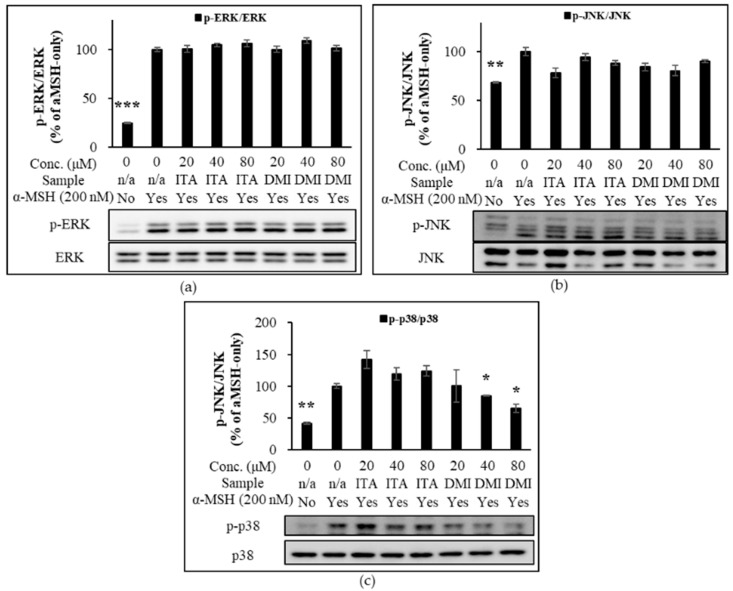
Effect of ITA and DMI on the α-MSH-induced expression of MAPK proteins. The expression level of p-ERK, p-JNK, and p-p38 was respectively normalized to that of total ERK, JNK, and p38. The relative expression level was compared to that of α-MSH only. (**a**) Effect of ITA and DMI on the α-MSH-induced phosphorylation of ERK. (**b**) Effect of ITA and DMI on the phosphorylation of JNK. (**c**) Effect of ITA and DMI on the α-MSH-induced expression of p38. The data were presented as the mean ± standard deviation of three independent experiments; * *p* < 0.05, ** *p* < 0.01, *** *p* < 0.005 compared with α-MSH only.

**Figure 6 molecules-27-04183-f006:**
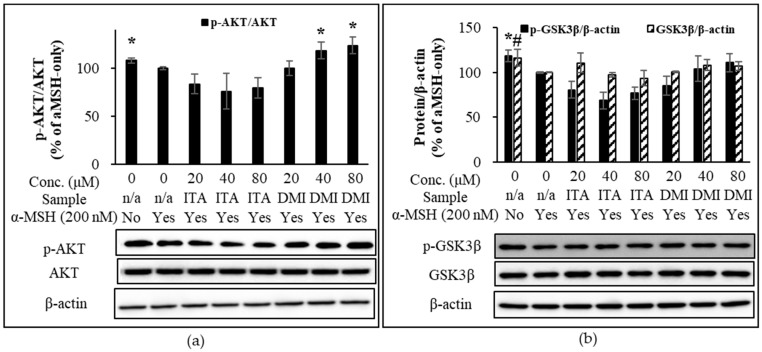
Effect of ITA and DMI on the α-MSH-induced phosphorylation of AKT and GSK3β. The expression level of p-AKT was normalized to that of total AKT while the expression level of p-GSK3β, GSK3β, and MITF was normalized to those of β-actin. The relative expression level was compared to that of α-MSH only. (**a**) Effect of ITA and DMI on the α-MSH-induced phosphorylation of AKT. (**b**) Effect of ITA and DMI on the α-MSH-induced expression of p-GSK3β and GSK3β at 15 min post-α-MSH treatment. The data were presented as the mean ± standard deviation of three independent experiments; * *p* < 0.05, ^#^ *p* < 0.05 compared with α-MSH only.

**Figure 7 molecules-27-04183-f007:**
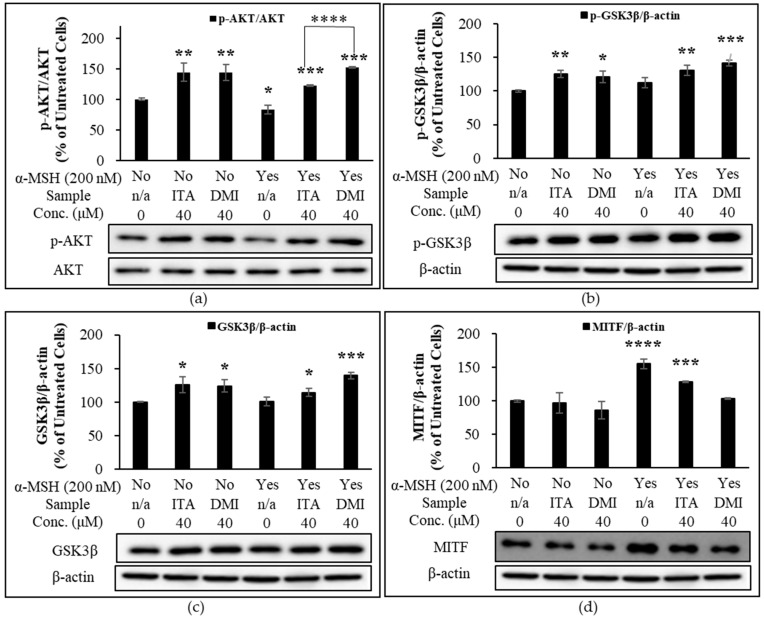
Effect of ITA and DMI on the α-MSH-induced expression of p-AKT, AKT, MITF, p-GSK3β, and GSK3β. B16F10 cells were treated in the presence or absence of α-MSH in combination with ITA or DMI (40 μM) for 6 h. Proteins were detected by immunoblotting. The expression level of p-AKT was normalized to that of total AKT while the expression levels of p-GSK3β, GSK3β, and MITF were normalized to those of β-actin. The relative expression level of each protein was compared to the basal expression level in untreated or α-MSH-stimulated cells. (**a**) Effect of ITA and DMI on the phosphorylation of AKT in the presence or absence of α-MSH. (**b**) Effect of ITA and DMI on the expression of MITF in the presence or absence of α-MSH. (**c**) Effect of ITA and DMI on the expression of p-GSK3β in the presence or absence of α-MSH. (**d**) Effect of ITA and DMI on the expression of GSK3β in the presence or absence of α-MSH. The data were presented as the mean ± standard deviation of three independent experiments; * *p* < 0.05, ** *p* < 0.01, *** *p* < 0.005, **** *p* < 0.001 compared with untreated cells.

## Data Availability

Not applicable.

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
