# Peer review of "Dimethyl Itaconate Reduces α-MSH-Induced Pigmentation via Modulation of AKT and p38 MAPK Signaling Pathways in B16F10 Mouse Melanoma Cells"

_molecules, 2022, doi:10.3390/molecules27134183_

Round 1
Reviewer 1 Report
This work studied the time-dependent effects of itaconic acid and dimethyl itaconate on α-MSH-induced melanogenesis in B16F10 cells. These cells were incubated with α-MSH alone or co-treated with itaconic acid or dimethyl itaconate. Dimethyl itaconate alone and α-MSH and dimethyl itaconate co-treatment showed a better inhibitory effect on MITF expression. The results indicate that dimethyl itaconate exerts antimelanogenic effect via modulation of the p38 mitogen-activated protein kinase and AKT/GSK3β signaling pathways.
- In Figure 1A, the melanin content of 40 μM itaconic acid was lower than that of 20 and 80 μM, however, the tyrosinase activity was not significantly inhibited after itaconic acid treated. The results were confused and it must recheck.
- Similar problem of the results was present in melanin content and tyrosinase activity dimethyl itaconate. The results must recheck and redo. In addition, the inhibition of melanin content may due to the cell death of 80 μM dimethyl itaconate treatment. The results must explain clearly in the manuscript.
- In Figure 3, the time course of all the protein expressions must explain clearly. Especially, the upstream and downstream regulation.
- In Figure 4a, the protein expression of tyrosinase was higher than α-MSH group, however, lower than α-MSH group at higher concentrations. The results must recheck and explain clearly in the text.
- The results of statistical analysis must label on all the Figures.
- This study presents many results of the protein expressions, however, the regulation between these proteins were unclear.
- All the results have to discuss and compare with other studies, and present in “Discussion” of the manuscript.
Author Response
This work studied the time-dependent effects of itaconic acid and dimethyl itaconate on α-MSH-induced melanogenesis in B16F10 cells. These cells were incubated with α-MSH alone or co-treated with itaconic acid or dimethyl itaconate. Dimethyl itaconate alone and α-MSH and dimethyl itaconate co-treatment showed a better inhibitory effect on MITF expression. The results indicate that dimethyl itaconate exerts antimelanogenic effect via modulation of the p38 mitogen-activated protein kinase and AKT/GSK3β signaling pathways.
Dear Reviewer,
The manuscript receive English editing service from Editage (KRHDH_16_02052022-0714919)
- In Figure 1A, the melanin content of 40 μM itaconic acid was lower than that of 20 and 80 μM, however, the tyrosinase activity was not significantly inhibited after itaconic acid treated. The results were confused and it must recheck.
The typos were fixed.
- Similar problem of the results was present in melanin content and tyrosinase activity dimethyl itaconate. The results must recheck and redo.
- Melanogenesis proteins including tyrosinase, tyrosinase-related protein 1 (TRP1) and TRP-2 cooperate for melanin synthesis. Also, tyrosinase substrates, co-factors, and cellular environment controls its catalytic activity [1].
- Tyrosinase activity could be different from melanin expression level [2].
In addition, the inhibition of melanin content may due to the cell death of 80 μM dimethyl itaconate treatment. The results must explain clearly in the manuscript.
The expression level of MITF, a transcription factor, was detected at 48 h post-sample treatment. Unlike the previous experiments in B16F10 cells were treated with sample compounds for 48 hours, B16F10 cells were treated for 72 hours to detect of the protein expression of TYR, TRP-1, TRP-2 proteins were detected at 72 h post-sample treatment.
Jeong et al. [2] investigated the anti-melanogenic effects of ethanol extracts of the leaves and roots of Patrinia villosa (Thunb.). For cell viability and melanin assay, they treated B16F10 cells with two different compounds (A and B) for 24 hours and examined cytotoxic effect of the compounds. The study was conducted with two concentrations of A and three different concentrations of B. At 24-hour post-sample treatment, B showed less than 90% cell viability. In another study, Hsu et al. [3] treated B16 cells with Lotus Seedp1od extract for 48 hours and examined cell viability. The relative cell viability (about 70%) at one concentration which is significantly low (p <0.01) compared to the untreated cells.
In another study [4], when B16F10 cells were treated with thymoquinone for 48 hours, the concentration at which cell viability just over 60% was used for melanin assay.
Depending on the assay results for cell viability and melanin content, the sample treatment concentration range would be determined [5].
References
[1] Mikami, M.; Sonoki, T.; Ito, M.; Funasaka, Y.; Suzuki, T.; Katagata, Y. Glycosylation of tyrosinase is a determinant of melanin production in cultured melanoma cells. 2013. 8, 1791-2997
[2] Chung, Y.; Hyun, C.-G. Inhibitory Effects of Pinostilbene Hydrate on Melanogenesis in B16F10 Melanoma Cells via ERK and p38 Signaling Pathways. International Journal of Molecular Sciences. 2020. 21, 4732
[3] Jeong, D.; Park, S. H.; Kim, M.-H.; Lee, S.; Cho, Y. K.; Kim, Y.A.; Park, B.J.; Lee, J.; Kang, H.; Cho, J. Y. Anti-melanogenic effects of ethanol extracts of the leaves and roots of Patrinia villosa (Thunb.) Juss through Their Inhibition of CREB and Induction of ERK and Autophagy, Molecules 2020, 25, 5375.
[4] Hsu, J.-Y.; Lin, H.H.; Li, T.-S.; Tseng, C.-Y.; Wong, Y.; Chen, J.-H. Anti-melanogenesis effects of Lotus Seedpod in vitro and in vivo. Nutrients 2020. 12, 3535.
[5] Jeong, H.; Yu, S.-M.; Kim, S.J. Inhibitory effects on melanogenesis by thymoquinone are mediated through β-catenin pathway in B16F10 mouse melanoma cells. Int. J. of Onc. 2020. 56, 379-389.
[6] Chung, S.; Lim, G.J.; Lee, J.Y. Quantitative analysis of melanin content in a three-dimensional melanoma cell culture. Sci. Rep. 2019. 9, 780.
- In Figure 3, the time course of all the protein expressions must explain clearly. Especially, the upstream and downstream regulation.
Figure 4 and section 2.4. would give clear explanation about the regulation of MAPK and GSK3β signaling pathways.
- In Figure 4a, the protein expression of tyrosinase was higher than α-MSH group, however, lower than α-MSH group at higher concentrations. The results must recheck and explain clearly in the text.
Unlike the previous experiments in which B16F10 cells were cultured with sample compounds for 48 hours, B16F10 cells were treated and incubated for 72 hours for the detection of tyrosinase, TRP-1, and TRP-2. This incubation time was previously used to quantify melanin content and cell viability. Accordingly, the manuscript was updated.
- The results of statistical analysis must label on all the Figures.
Fixed.
- This study presents many results of the protein expressions, however, the regulation between these proteins were unclear.
We scrutinized our previous manuscript and removed irrelevant contents and put more relevant information.
- All the results have to discuss and compare with other studies, and present in “Discussion” of the manuscript.
I revised my old manuscript. I wish this updated manuscript would give more solid conclusion.

Reviewer 2 Report
This is an interesting paper, but to the proper interpretation of the results, some important modifications must be undertaken.
1. Please accompany the text with a clear scheme of regulation, as the description contained in the “Introduction” is very difficult to follow and to dissect
2. My primary doubt concerning the experimental part of the paper is – how much do the authors observe DMI affecting the pigmentation of B16, and how much the a-MSH-induced pigmentation. This aspect must be particularly paid the attention to, and the observations may be repeated on normel melanocytes, because B16 reveals, as many melanomas, impairment in pigmentation.
3. Line 216 and further. Melanogenesis is primarily the pathway of metamorphosis of DOPA-quinone, which easily progresses non-enzymatically. Enzymes only modify the pre-existing process. And DOPA is NOT an intermediate of the hydroxylation of L-Tyr by tyrosinase. DOPA is not a product of tyrosinase activity, only it may be a substrate, and it is produced in the Raper-Masson pathway in further steps, non-enzymatically, via “redox exchange” reaction. There may be DOPA produced in different concentrations by another enzyme – tyrosine hydroxylase, to make it able to start melanogenesis, at all. DOPA is necessary to activate tyrosinase (to reduce met-tyrosinase) and must be present to set the machinery in motion. So, it is not true that tyrosinase produces dopaquinone VIA DOPA, the paper must be modified and the results re-interpreted. To become familiar with these facts, please read and cite: Schallreuter et al. EXD, doi: 10.1111/j.1600-0625.2007.00675.x, and the most important; Land et al., Methods in enzymology doi: 10.1016/S0076-6879(04)78005-2. This idea was used in many other items, including e.g. Plonka et al., doi: 10.1111/j.1365-2133.2006.07376.x
4. I have a serious doubts concerning the method of determining tyrosinase activity. First of all, tyrosinase possesses two enzymatic activities (besides the met-tyrosinase activation). – the cresolase activity (oxidation of tyrosine to dopaquinone), and catecholase (oxidation of DOPA to dopaquinone, DOPA generated via redox exchange or by tyrosine hydroxylase).
5. Therefore, the interpretation of tyrosinase activity is difficult and troublesome. Just monitoring OD at 490 nm when, arbitrarily determined solution color change is very rough. Please see Winder & Harris 1991 doi: 10.1111/j.1432-1033.1991.tb16018.x and further papers.
6. Please carefully correct all the typos and technical errors. Here I only list the numbers of erroneous lines: 68, 325, 347, through the text – please replace, when necessary, the small letter “x” with the “times” symbol ×, please use lowercase 2 in the chemical formula of CO2, use uppercase numbers when indicating powers, 389, 405 (symbol for “hours” is “h”), 461.
Author Response
This is an interesting paper, but to the proper interpretation of the results, some important modifications must be undertaken.
Dear Reviewer,
The manuscript receive English editing service from Editage (KRHDH_16_02052022-0714919)
1. Please accompany the text with a clear scheme of regulation, as the description contained in the “Introduction” is very difficult to follow and to dissect
- We securitized our previous manuscript and removed irrelevant contents and put more relevant information.
2. My primary doubt concerning the experimental part of the paper is – how much do the authors observe DMI affecting the pigmentation of B16, and how much the a-MSH-induced pigmentation. This aspect must be particularly paid the attention to, and the observations may be repeated on normel melanocytes, because B16 reveals, as many melanomas, impairment in pigmentation.
- Melanin assay and cell viability assay were repeated at least four times independently. The data was obtained by three independent experiments.
- Our lab has B16F10 cells only for pigmentation test.
3. Line 216 and further. Melanogenesis is primarily the pathway of metamorphosis of DOPA-quinone, which easily progresses non-enzymatically. Enzymes only modify the pre-existing process. And DOPA is NOT an intermediate of the hydroxylation of L-Tyr by tyrosinase. DOPA is not a product of tyrosinase activity, only it may be a substrate, and it is produced in the Raper-Masson pathway in further steps, non-enzymatically, via “redox exchange” reaction. There may be DOPA produced in different concentrations by another enzyme – tyrosine hydroxylase, to make it able to start melanogenesis, at all. DOPA is necessary to activate tyrosinase (to reduce met-tyrosinase) and must be present to set the machinery in motion. So, it is not true that tyrosinase produces dopaquinone VIA DOPA, the paper must be modified and the results re-interpreted. To become familiar with these facts, please read and cite: Schallreuter et al. EXD, doi: 10.1111/j.1600-0625.2007.00675.x, and the most important; Land et al., Methods in enzymology doi: 10.1016/S0076-6879(04)78005-2. This idea was used in many other items, including e.g. Plonka et al., doi: 10.1111/j.1365-2133.2006.07376.x
Dear Reviewer,
I deeply appreciate you for the information. I would to caution to say that our lab followed the methods in research papers as shown below.
[1] Han, H.; Kim, Y.; Mo, H.; Choi, S. H.; Lee, K.; Rim, Y. A.; Ju, J. H. Preferential stimulation of melanocyte by M2 macrophages to produce melanin through vascular endothelial growth factor. Scientific Reports. 2022. 12, 6416
[2] Jeon, S.; Kim, N.-H.; Koo, B.-S.; Kim, J.-Y.; Lee, A.-Y. Lotus (Nelumbo nuficer) flower essential oil increased melanogenesis in normal human melanocytes. 2009. 41, 517-524.
[3] Jung, G.-D.; Yang, J.-Y.; Song, E.-S.; Park, J.-W. Stimulation of melnogenesis by glycyrrhizin in B16 melanoma cells. Experimental and Molecular Medicine, 2001. 33,131-135
4. I have a serious doubts concerning the method of determining tyrosinase activity. First of all, -tyrosinase possesses two enzymatic activities (besides the met-tyrosinase activation). – the cresolase activity (oxidation of tyrosine to dopaquinone), and catecholase (oxidation of DOPA to dopaquinone, DOPA generated via redox exchange or by tyrosine hydroxylase).
- Same as above.
5. Therefore, the interpretation of tyrosinase activity is difficult and troublesome. Just monitoring OD at 490 nm when, arbitrarily determined solution color change is very rough. Please see Winder & Harris 1991 doi: 10.1111/j.1432-1033.1991.tb16018.x and further papers.
- Same as above.
6. Please carefully correct all the typos and technical errors. Here I only list the numbers of erroneous lines: 68, 325, 347, through the text – please replace, when necessary, the small letter “x” with the “times” symbol ×, please use lowercase 2 in the chemical formula of CO2, use uppercase numbers when indicating powers, 389, 405 (symbol for “hours” is “h”), 461.F
- Fixed-

Reviewer 3 Report
Authors propose that ITA and DMI reduce pigmentation in B16F10 cells. Hypothesis is: ITA and or DIMI induces NRF2, which suppresses melanin production.
There are major issues in this manuscript
1.) Figure 1 b ITA 20 and 80 microM do not reduce melanin content compared to 0 microM. Also DMI 40 micro shows nearly no suppression.
-How do authors conclude that ITA reduces melanin when only 40 microM shows an effect?
-Why is DMI 40microM higher than 20 and 80 microM?
Figure 1c shows that cell viability decreases with 80microM DMI. This needs to be taken serious when interpreting subsequent data (e.g.: Figure 2 only 80microM DMI shows decrease in tyrosinase activity)!
2.) Figure 3.
Wrong labelling 3a is p-AKt not p-Erk, as written in legends.
X axis, 0, 15, 30 ,60 is labelled as microM, but described as a time-course !?
3.) Figure 4
-MITF + ITA and TRP-1 and TRP-2 + ITA and DMI does not decline (except for 80 microM, which is toxic)!
-Also, Figure 1 shows melanin content decrease at 20 microM DMI, but Figure 4 shows an increase at 20 microM.
4.) Figure 5
-No clear conclusion possible
5.) Figure 6
-Signaling data not convincing. After 24 hours MITF downregulation not visible.
Hence, title and conclusion are not supported by strong data.
Minor points:
The statistical analysis section needs to be improved
Author Response
Authors propose that ITA and DMI reduce pigmentation in B16F10 cells. Hypothesis is: ITA and or DIMI induces NRF2, which suppresses melanin production.
There are major issues in this manuscript
Dear Reviewer,
The manuscript receive English editing service from Editage (KRHDH_16_02052022-0714919)
1.) Figure 1 b ITA 20 and 80 microM do not reduce melanin content compared to 0 microM. Also DMI 40 micro shows nearly no suppression.
-How do authors conclude that ITA reduces melanin when only 40 microM shows an effect?
The typo was corrected.
-Why is DMI 40microM higher than 20 and 80 microM?
The typo was corrected.
Figure 1c shows that cell viability decreases with 80microM DMI. This needs to be taken serious when interpreting subsequent data (e.g.: Figure 2 only 80microM DMI shows decrease in tyrosinase activity)!
Unlike the previous experiments in B16F10 cells were treated with sample compounds for 48 hours, B16F10 cells were treated for 72 hours to detect of the protein expression of TYR, TRP-1, TRP-2 proteins were detected at 72 h post-sample treatment.
Jeong et al. [1] investigated the anti-melanogenic effects of ethanol extracts of the leaves and roots of Patrinia villosa (Thunb.). For cell viability and melanin assay, they treated B16F10 cells with two different compounds (A and B) for 24 hours and examined cytotoxic effect of the compounds. The study was conducted with two concentrations of Compound A and three different concentrations of Compound B. At 24-hour post-sample treatment, Compound B showed less than 90% cell viability. In another study, Hsu et al. [2] treated B16 cells with Lotus Seedp1od extract for 48 hours and examined cell viability. The relative cell viability (about 70%) at one concentration which is significantly low (p <0.01) compared to the untreated cells.
In another study [3], when B16F10 cells were treated with thymoquinone for 48 hours, the concentration at which cell viability just over 60% was used for melanin assay [3].
Depending on the assay results for cell viability and melanin content, the sample treatment concentration range would be determined [4].
References
[1] Jeong, D.; Park, S. H.; Kim, M.-H.; Lee, S.; Cho, Y. K.; Kim, Y.A.; Park, B.J.; Lee, J.; Kang, H.; Cho, J. Y. Anti-melanogenic effects of ethanol extracts of the leaves and roots of Patrinia villosa (Thunb.) Juss through Their Inhibition of CREB and Induction of ERK and Autophagy, Molecules 2020, 25, 5375.
[2] Hsu, J.-Y.; Lin, H.H.; Li, T.-S.; Tseng, C.-Y.; Wong, Y.; Chen, J.-H. Anti-melanogenesis effects of Lotus Seedpod in vitro and in vivo. Nutrients 2020. 12, 3535.
[3] Jeong, H.; Yu, S.-M.; Kim, S.J. Inhibitory effects on melanogenesis by thymoquinone are mediated through β-catenin pathway in B16F10 mouse melanoma cells. Int. J. of Onc. 2020. 56, 379-389.
[4] Chung, S.; Lim, G.J.; Lee, J.Y. Quantitative analysis of melanin content in a three-dimensional melanoma cell culture. Sci. Rep. 2019. 9, 780.
2.) Figure 3.
Wrong labelling 3a is p-AKt not p-Erk, as written in legends.
Fixed.
X axis, 0, 15, 30 ,60 is labelled as microM, but described as a time-course !?
Fixed.
3.) Figure 4
-MITF + ITA and TRP-1 and TRP-2 + ITA and DMI does not decline (except for 80 microM, which is toxic)!
The expression level of MITF, a transcription factor, was detected at 48 h post-sample treatment. Unlike the previous experiments in B16F10 cells were treated with sample compounds for 48 hours, B16F10 cells were treated for 72 hours to detect of the protein expression of TYR, TRP-1, TRP-2 proteins were detected at 72 h post-sample treatment.
-Also, Figure 1 shows melanin content decrease at 20 microM DMI, but Figure 4 shows an increase at 20 microM.
This typo was corrected.
4.) Figure 5
-No clear conclusion possible
I wish that the revised manuscript would give more solid conclusion.
5.) Figure 6
-Signaling data not convincing. After 24 hours MITF downregulation not visible.
Hence, title and conclusion are not supported by strong data.
At 24 hour post-sample treatment, MITF was slightly increased in α-MSH only relative to untreated cell. This figure intended to show that itaconic acid increased MITF expression while dimethyl itaconate decrease MITF expression to the basal expression level in untreated cells. Since this 24 hour data provided insufficient evident, we edited out. However, I revised my old manuscript. I wish this updated manuscript would give more solid conclusion.
Minor points:
The statistical analysis section needs to be improved,
The statistical analysis section was revised and each caption for the figures contains the statistical process.

Round 2
Reviewer 2 Report
I must maintain my previous evaluation. The Authors devoted a lot of time to re-write the paper but they did not address the majority of my comments. They still maintain that Tyrosinase oxidises L-Tyr to L-DOPA, and cite the papers that are already cited in the paper itself, which I noticed on the first round. It is NOT the response to the issues, nor the expected revision. I find that improving the factual data and references and mentioning the requested issues in the Discussion would be enough. I cannot reject this paper as the Authors have not spoiled the paper, just maintain its status quo, and my primary decision is "major revision" but as the review will be published along with the paper, I cannot accept this version. Moreover, in the cited papers, there is no information which entitles to maintain their opinion and, still moreover, the cited Jung et al 2001 have themselves written about the catecholase (DOPA-oxidation) activity of the enzyme.
Author Response
I must maintain my previous evaluation. The Authors devoted a lot of time to re-write the paper but they did not address the majority of my comments. They still maintain that Tyrosinase oxidises L-Tyr to L-DOPA, and cite the papers that are already cited in the paper itself, which I noticed on the first round. It is NOT the response to the issues, nor the expected revision. I find that improving the factual data and references and mentioning the requested issues in the Discussion would be enough. I cannot reject this paper as the Authors have not spoiled the paper, just maintain its status quo, and my primary decision is "major revision" but as the review will be published along with the paper, I cannot accept this version. Moreover, in the cited papers, there is no information which entitles to maintain their opinion and, still moreover, the cited Jung et al 2001 have themselves written about the catecholase (DOPA-oxidation) activity of the enzyme.
The concept of L-DOPA oxidation and your suggestions were all considered in the text. Please see Line 111 – 125 and Line 347 – 351 and Line 408 – 423. Figure 2 were accordingly modified. In “L-DOPA Oxidation” section, I cited a reference [58]. In this article, Chen et al. presented “UV-VIS spectra for the oxidation of L-DOPA” (Figure 8). The wavelength 490 nm is near to the maximum of the spectra.

Reviewer 3 Report
Authors have made changes to the manuscript. Results are now presented better.
Is AKT driving pigmentation, can p38 prevent it? This needs to be explained in the text.
Author Response
Authors have made changes to the manuscript. Results are now presented better.
Is AKT driving pigmentation, can p38 prevent it? This needs to be explained in the text.
Your comments were considered. Constitutively active mutant of AKT inhibits melanogenesis. [Reference 48] For the details, please see Line 164 – 170 and Line 343 – 347.

Round 3
Reviewer 2 Report
The correcttions introduced by the authors are now enough to accept this manuscript.